# Regulated degradation of the inner nuclear membrane protein SUN2 maintains nuclear envelope architecture and function

Logesvaran Krshnan[1†], Wingyan Skyla Siu[1], Michael Van de Weijer[1], Daniel Hayward[1], Elena Navarro Guerrero[2], Ulrike Gruneberg[1], Pedro Carvalho[1*]

[1]Sir William Dunn School of Pathology, University of Oxford, Oxford, United Kingdom; [2]Nuffield Department of Medicine, Target Discovery Institute, University of Oxford, Oxford, United Kingdom

**Abstract** Nuclear architecture and functions depend on dynamic interactions between nuclear components (such as chromatin) and inner nuclear membrane (INM) proteins. Mutations in INM proteins interfering with these interactions result in disease. However, mechanisms controlling the levels and turnover of INM proteins remain unknown. Here, we describe a mechanism of regulated degradation of the INM SUN domain-containing protein 2 (SUN2). We show that Casein Kinase 2 and the C-terminal domain Nuclear Envelope Phosphatase 1 (CTDNEP1) have opposing effects on SUN2 levels by regulating SUN2 binding to the ubiquitin ligase Skp/Cullin1/F-Box$^{\beta TrCP}$ (SCF$^{\beta TrCP}$). Upon binding to phosphorylated SUN2, SCF$^{\beta TrCP}$ promotes its ubiquitination. Ubiquitinated SUN2 is membrane extracted by the AAA ATPase p97 and delivered to the proteasome for degradation. Importantly, accumulation of non-degradable SUN2 results in aberrant nuclear architecture, vulnerability to DNA damage and increased lagging chromosomes in mitosis. These findings uncover a central role of proteolysis in INM protein homeostasis.

**\*For correspondence:**
pedro.carvalho@path.ox.ac.uk

**Present address:** [†]Medical Research Council Protein Phosphorylation and Ubiquitylation Unit (MRC-PPU), School of Life Sciences, University of Dundee, Dundee, United Kingdom

**Competing interest:** The authors declare that no competing interests exist.

## Editor's evaluation

This important paper describes a "ERAD-like" pathway for turnover of the SUN2 protein. Here, ubiquitylation of SUN2 in the nucleoplasm by the SCFbTRCP ubiquitin ligase leads to extraction of the membrane protein by p97 for delivery to the proteasome in a process that is regulated by the CK2 kinase and the CTDNEP1 phosphatase. Non-degradable forms of SUN2 promote altered nuclear architecture and a delay in double strand break repair. The conclusions are based on strong biochemical and cell biological data, and are supported by multiple types of genetic approaches.

## Introduction

The organization of eukaryotic chromosomes within the cell nucleus depends on regulated and dynamic interactions with the nuclear envelope (NE). These interactions influence a myriad of cellular processes, from gene regulation and repair to cell motility and fate (*Mekhail and Moazed, 2010*). Chromosome interactions involve primarily proteins at the inner nuclear membrane (INM), which together with the outer nuclear membrane (ONM), forms the NE. While continuous with the remaining Endoplasmic Reticulum (ER) membrane, the INM has a distinct set of proteins (*Pawar and Kutay, 2021*). Mutations in INM proteins are associated with a wide range of diseases, such as muscular

dystrophies and premature ageing syndromes, underscoring their importance for nuclear architecture and function.

Among the diverse INM proteome, Sad1p/UNC84 (SUN) domain-containing proteins SUN1 and SUN2 are unique as they couple nuclear and cytoplasmic events via LINC (Linker of nucleoskeleton and cytoskeleton) complexes (*Crisp et al., 2006*; *Hodzic et al., 2004*; *Padmakumar et al., 2005*). These molecular bridges across the NE are mediated by interactions of the luminal domains of SUN1 or SUN2 homotrimers in the INM and Nesprin proteins at the ONM (*Crisp et al., 2006*; *Padmakumar et al., 2005*; *Sosa et al., 2012*). Other domains of SUN and Nesprin proteins engage with various components in the nucleus and the cytoskeleton, respectively, conferring the LINC complex with diverse functions. Indeed, by transducing forces from the cytoskeleton into nucleus, SUN1 and SUN2 have been implicated in controlling nuclear positioning within the cell (*Lei et al., 2009*; *Zhang et al., 2009*), NE anchoring of specific chromatin loci, facilitating NE disassembly during mitosis (*Turgay et al., 2014*) and DNA repair (*Lei et al., 2012*; *Lottersberger et al., 2015*). SUN proteins have also been implicated in HIV replication and trafficking within host cells (*Donahue et al., 2016*). In these various functions, SUN1 and SUN2 appear to play both redundant and unique roles. However, our understanding of SUN proteins and the mechanisms by which they contribute to nuclear organization are far from complete.

Recent work in budding yeast showed that protein homeostasis at the INM relies on ubiquitin-dependent protein degradation (*Foresti et al., 2014*; *Khmelinskii et al., 2014*). This process is mainly mediated by an INM localized ubiquitin ligase complex, the Asi complex, which promotes ubiquitination of damaged and mislocalized membrane proteins (*Foresti et al., 2014*; *Khmelinskii et al., 2014*; *Natarajan et al., 2020*). Proteins ubiquitinated by the Asi complex are subsequently extracted from the INM into the nucleoplasm by a conserved ATPase complex called Cdc48 in yeast and p97 in mammals and delivered to the proteasome for degradation. Mechanistically, Asi-mediated protein degradation is similar to the ER-associated protein degradation (ERAD), a vital quality control process in ER membranes exposed to the cytosol (*Christianson and Carvalho, 2022*; *Wu and Rapoport, 2018*). In fact, the yeast ERAD ubiquitin ligase Doa10 also localizes to the INM and degrades both nucleoplasmic and INM proteins (*Boban et al., 2014*; *Deng and Hochstrasser, 2006*; *Swanson et al., 2001*). Despite the conservation of the ERAD process across all eukaryotes, sequence-based homologues of the Asi complex appear to be circumscribed to certain fungi.

While a functional counterpart of the yeast Asi complex has not been identified in higher eukaryotes, there are hints that the ubiquitin system influences protein homeostasis at the INM in mammals. Disease-associated mutations in the Lamin-B receptor (LBR) appear to be removed from the INM through a process involving ubiquitination (*Tsai et al., 2016*). Perturbations of NE homeostasis, such as mutations in Torsin ATPase, result in INM herniations enriched in polyubiquitin conjugates (*Laudermilch et al., 2016*). Proteomics studies suggest links between the Skp/Cullin1/F-Box$^{\beta\text{-TrCP1/2}}$ (SCF$^{\beta\text{-TrCP1/2}}$) ubiquitin ligase complex and the INM protein SUN2 (*Coyaud et al., 2015*; *Huttlin et al., 2021*; *Huttlin et al., 2017*; *Loveless et al., 2015*; *Low et al., 2014*; *Sowa et al., 2009*). The INM protein Emerin has also been shown to interact with ubiquitin ligases (*Khanna et al., 2018*). However, the significance of these observations and the overall role of protein degradation to INM protein homeostasis remain largely unexplored.

Here, we show that SUN2 levels at the INM are controlled by an ERAD-like mechanism. This process involves SUN2 ubiquitination by the SCF$^{\beta\text{TrCP}}$ ubiquitin ligase, membrane extraction by the p97 ATPase complex, and degradation by the proteasome. We further show that the opposing activities of Casein Kinase 2 (CK2) and the C-terminal domain Nuclear Envelope Phosphatase 1 (CTDNEP1) regulate SUN2 degradation. This kinase/phosphatase balance regulates SCF$^{\beta\text{TrCP}}$ binding to a non-canonical recognition motif in the SUN2 nucleoplasmic domain that is required for subsequent SUN2 ubiquitination. Finally, we show that accumulation of non-degradable SUN2 results in defects in NE architecture that impact nuclear functions.

## Results

### SUN2 stability depends on a non-canonical recognition motif for the SCF$^{\beta TrCP}$ ubiquitin ligase

The SCF$^{\beta TrCP}$ ubiquitin ligase promotes ubiquitination and degradation of substrates containing defined recognition motifs (*Frescas and Pagano, 2008*). To study the relation between SUN2 and SCF$^{\beta TrCP}$, we searched the SUN2 nucleoplasmic region for potential SCF$^{\beta TrCP}$-binding motifs. While the canonical DSGXXS recognition motif is absent, SUN2 contains two regions with related amino acid sequences, hereafter called Sites 1 and 2, respectively. Importantly, Sites 1 and 2 are conserved across vertebrates and are absent in SUN1, a SUN2 homologue (*Figure 1—figure supplement 1A–C*).

Binding of SCF$^{\beta TrCP}$ to canonical and non-canonical sites on substrates requires their prior phosphorylation at two consensus serine residues (*Frescas and Pagano, 2008*). To test the potential involvement of Sites 1 and 2 in SUN2 degradation, the critical consensus serine residues were mutated to alanine to abolish phosphorylation, and to aspartate to mimic constitutive phosphorylation (*Figure 1A*). Wild-type SUN2 (hereafter called SUN2 WT) as well as Site 1 and 2 mutants were stably integrated at the single flippase recognition target (FRT) site of HEK293 TRex Flp-In (HEK TF) cell lines under a doxy-cycline/tetracycline-inducible promoter. The various SUN2 constructs were expressed as C-terminal fusions to superfolder green fluorescent protein (sfGFP) and hemagglutinin (HA) tags for easy detection (*Figure 1B*). Upon doxycycline (Dox) induction, there was robust detection of SUN2 WT by flow cytometry. Alanine mutations in Site 1 or 2 (Site 1$^A$ or 2$^A$, respectively) resulted in even higher steady-state SUN2 levels (*Figure 1C*). This increase was particularly noticeable for Site 2$^A$ mutant and was further increased by combining alanine mutations in both sites. Conversely, phosphomimetic aspartate mutations at Sites 1 and 2 (Sites 1$^D$ and 2$^D$, respectively) resulted in much lower SUN2 steady-state levels, with the Site 2$^D$ mutant displaying again a stronger effect (*Figure 1D*). Combination of aspartate mutations in both sites (Sites 1$^D$ and 2$^D$) resulted in further decrease in SUN2 steady-state levels, suggesting that phosphorylation at Sites 1 and 2 have an additive effect. Since all SUN2 derivatives are expressed from the same promoter, these data suggest that SUN2 stability is controlled by non-canonical SCF$^{\beta TrCP}$ recognition motifs, with Site 2 having a more prominent contribution.

To directly assess the effect of these mutations on SUN2 degradation, we performed chase experiments upon inhibition of protein translation with cycloheximide (CHX). We observed that SUN2 WT was degraded with a half-life of ~240 min (*Figure 1E*). Consistent with the analysis at steady state, SUN2 Site 2$^A$ mutant was stable while the phosphomimetic aspartate mutation dramatically accelerated SUN2 degradation. SUN2 degradation was further accelerated by combining aspartate mutations at Sites 1 and 2 (*Figure 1E*). Interestingly, turnover of endogenous SUN2 was influenced and mirrored the turnover of the various transgenes (*Figure 1E*). Given that SUN2 functions as a trimer (*Sosa et al., 2012*), this observation likely reflects assembly of endogenous and transgenic molecules into trimeric SUN2 complexes. In contrast, the SUN2 homologue SUN1 was stable under all the conditions tested. A small but reproducible increase in SUN1 steady-state levels was observed in cells expressing unstable SUN2 variants. These data confirmed our initial observations and showed that SUN2 degradation is controlled by non-canonical SCF$^{\beta TrCP}$ recognition motifs.

### INM degradation of SUN2 by an ERAD-like process

To test the role of the ubiquitin ligase SCF$^{\beta TrCP}$ in SUN2 degradation, its activity was acutely blocked with MLN4924. This well-characterized small molecule inhibits Cullin RING ligases like SCF$^{\beta TrCP}$ by preventing their activation by neddylation (*Lottersberger et al., 2015*). Acute SCF$^{\beta TrCP}$ inhibition with MLN4924 resulted in the stabilization of both WT (*Figure 2A, B*) and Site 2$^D$ (*Figure 2A, C*) SUN2, whereas the steady-state levels of the stable SUN2 Site 2$^A$ mutant were unaffected (*Figure 2A*). Given that MLN4924 is a general inhibitor of Cullin RING ligases, the contribution of SCF$^{\beta TrCP}$ in SUN2 degradation was directly tested upon depletion of the F-Box proteins βTrCP1 and βTrCP2. Depletion of βTrCP1 did not affect the levels of SUN2 Site 2$^D$, while βTrCP2 depletion resulted in increased steady-state levels of this short-lived SUN2 mutant (*Figure 2—figure supplement 1A, B*). Further increase in SUN2 Site 2$^D$ levels was observed when βTrCP1 and βTrCP2 were depleted simultaneously showing that they are redundant in promoting SUN2 degradation. Depletion of other F-Box proteins, such as FBXO2 and FBXO6, had no effect on SUN2 Site 2$^D$ levels indicating that the effect observed for βTrCP1 and 2 is specific (*Figure 2—figure supplement 1A, B*). These data indicate that SCF$^{\beta TrCP}$ ubiquitin ligase promotes SUN2 degradation and highlight the importance of Site 2 in the process.

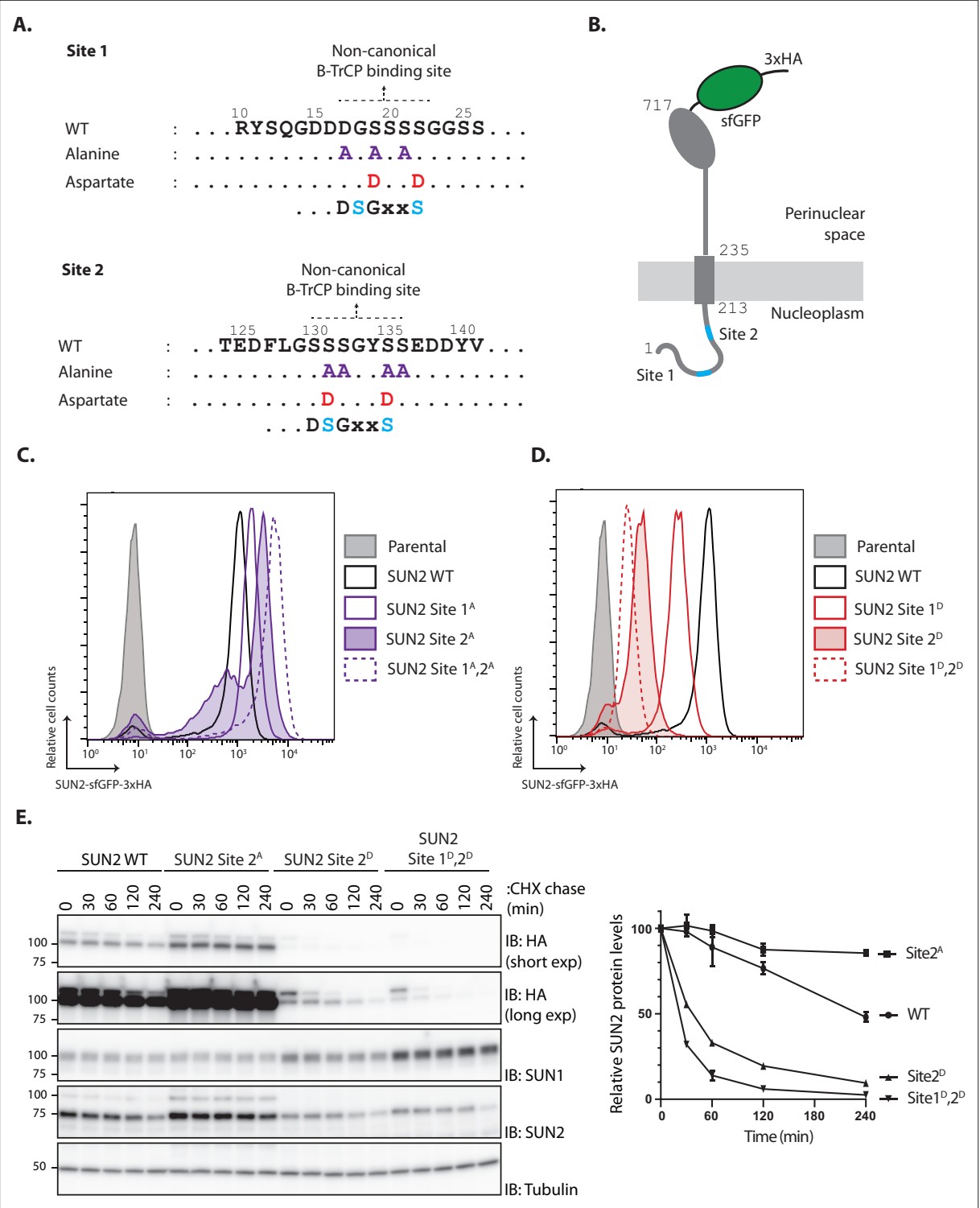

**Figure 1.** SUN2 stability depends on a non-canonical recognition motif for the SCF^βTrCP ubiquitin ligase. (**A**) SUN2 nucleoplasmic regions containing non-canonical recognition motifs for the SCF^βTrCP hereafter called Sites 1 and 2. In each case, the residues mutated to Alanine (purple) or to Aspartate (red) are indicated. The canonical SCF^βTrCP recognition motif (DSGXXS) is indicated. SCF^βTrCP binding occurs only to serine residues (Cyan) that are phosphorylated. (**B**) Schematic representation of SUN2-GFP-3HA construct used in most experiments. Sites 1 and 2 in the SUN2 nucleoplasmic region are indicated in Cyan. (**C, D**) Flow cytometry analysis of doxycycline-induced expression of SUN2 WT or derivatives in HEK TF cells. Analysis was performed 24 hr after SUN2 expression. (**E**) Analysis of the stability of SUN2 WT and the indicated mutants after inhibition of protein synthesis by cycloheximide (CHX). Cell extracts were analyzed by sodium dodecyl sulfate–polyacrylamide gel electrophoresis (SDS–PAGE) and immunoblotting.

*Figure 1 continued on next page*

Figure 1 continued

Transgenic WT and mutant SUN2 were detected with anti-HA antibody. Endogenous SUN2 was detected with anti-SUN2 antibody. This antibody recognizes the C-terminal peptide of SUN2 and is deficient in recognizing SUN2 if tagged C-terminally. Endogenous SUN1 was detected with anti-SUN1 antibody. Tubulin was used as a loading control and detected with an anti-Tubulin antibody.

The online version of this article includes the following source data and figure supplement(s) for figure 1:

**Source data 1.** File contains original immunoblots for *Figure 1E*.

**Figure supplement 1.** Evolutionary conservation analysis of SUN2 protein sequence.

Next, we tested the role of the p97 ATPase complex, which extracts ubiquitinated proteins from the membrane during ERAD (*Christianson and Carvalho, 2022*; *Wu and Rapoport, 2018*). Inhibition of p97 with CB-5803 also resulted in stabilization of both SUN2 WT (*Figure 2A*) and phosphomimetic mutant Site $2^D$ (*Figure 2A–C*). Moreover, SUN2 was stabilized upon inhibition of the proteasome with Bortezomib (*Figure 2A–C*), while inhibition of lysosomal protein degradation with Bafilomycin A had no effect on SUN2 levels (*Figure 2C*). Similar results were obtained when the levels of phosphomimetic mutant Site $2^D$ were analyzed by immunofluorescence, which showed accumulation at the NE (*Figure 2D*). Together, our data suggest SUN2 is degraded via an ERAD-like process involving the ubiquitin ligase SCF$^{\beta TrCP}$, the p97 ATPase complex, and the proteasome.

## Site 2 is critical for SUN2 recognition and ubiquitination by SCF$^{\beta TrCP}$

Analysis of protein degradation suggests a critical role of Site 2 in SUN2 recognition and ubiquitination by SCF$^{\beta TrCP}$. To directly analyze the contribution of Sites 1 and 2 in SUN2 binding to SCF$^{\beta TrCP}$, we used immunoprecipitation. We found that endogenous β-TrCP coprecipitated with SUN2 WT. SUN2 binding to β-TrCP was largely unaffected by Site $1^A$ and $1^D$ mutations. In contrast, β-TrCP binding to SUN2 was lost in cells expressing the Site $2^A$ mutant, which cannot be phosphorylated (*Figure 3A*). Conversely, the amount of β-TrCP coprecipitated by the phosphomimetic SUN2 Site $2^D$ was greatly increased, even if Site $2^D$ levels were much lower. The amount of β-TrCP coprecipitated with SUN2 WT and Site $2^D$ was further increased in cells treated with the p97 inhibitor CB-5083, suggesting that inhibition of SUN2 degradation stabilized its interaction with the SCF$^{\beta TrCP}$ ubiquitin ligase. Under these conditions, neddylated Cullin-1 also coprecipitated, indicating that SUN2 interacts primarily with the active pool of SCF$^{\beta TrCP}$ complex (*Figure 3A*).

Next, we used immunoprecipitation under denaturing conditions to analyze SUN2 ubiquitination. Ubiquitinated SUN2 WT was readily detected in unperturbed cells and the amount of ubiquitin conjugates increased in cells treated with inhibitors of p97 ATPase or the proteasome, which hinder protein retrotranslocation or degradation, respectively (*Figure 3B*). Mutations in Site 1 resulted in levels of ubiquitin conjugates comparable to SUN2 WT. Consistent with its fast turnover, Site $2^D$ mutant displayed increased ubiquitin conjugates, which became even more prominent upon p97 or proteasome inhibition. Importantly, ubiquitin conjugates to SUN2 WT and its derivatives were virtually undetectable upon inhibition of SCF$^{\beta TrCP}$ with MLN4924 (*Figure 3B*). Irrespective of the conditions used, ubiquitin conjugates were absent on SUN2 Site $2^A$ (*Figure 3B*). Thus, SUN2 ubiquitination correlates with its ability to interact with SCF$^{\beta TrCP}$. Altogether, these data indicate SUN2 Site 2 is a bona fide SCF$^{\beta TrCP}$-binding motif, recognized by βTrCP in a phosphorylation-dependent manner.

## Genome-wide screening identifies Casein Kinase 2 as a positive regulator of SUN2 degradation

To gain further insight on the mechanism of SUN2 degradation, we performed a genome-wide CRISPR-Cas9-based genetic screen using the short-lived SUN2 Site $2^D$ mutant as a reporter. HEK TF cells with the SUN2 Site $2^D$-sfGFP-HA transgene were transduced with the Toronto KnockOut (TKO) lentivirus library consisting of an average of 4 sgRNAs to each of the 18,053 human genes and additional control sgRNAs (*Hart et al., 2017*). Mutant cells with high levels of GFP (top 1%), suggestive of impaired degradation of SUN2 Site $2^D$-sfGFP-HA, were isolated by flow cytometry at days 9, 13, and 17 after transduction as described (*van de Weijer et al., 2020*). Analysis of multiple timepoints allows identification of mutations in essential genes, which manifest early but drop out from later timepoints (*Hart et al., 2015*). Next-generation sequencing was used to sequence and quantify sgRNAs in reference and sorted cell populations (*Figure 4A* and *Figure 4—figure supplement 1A*). Gene rankings

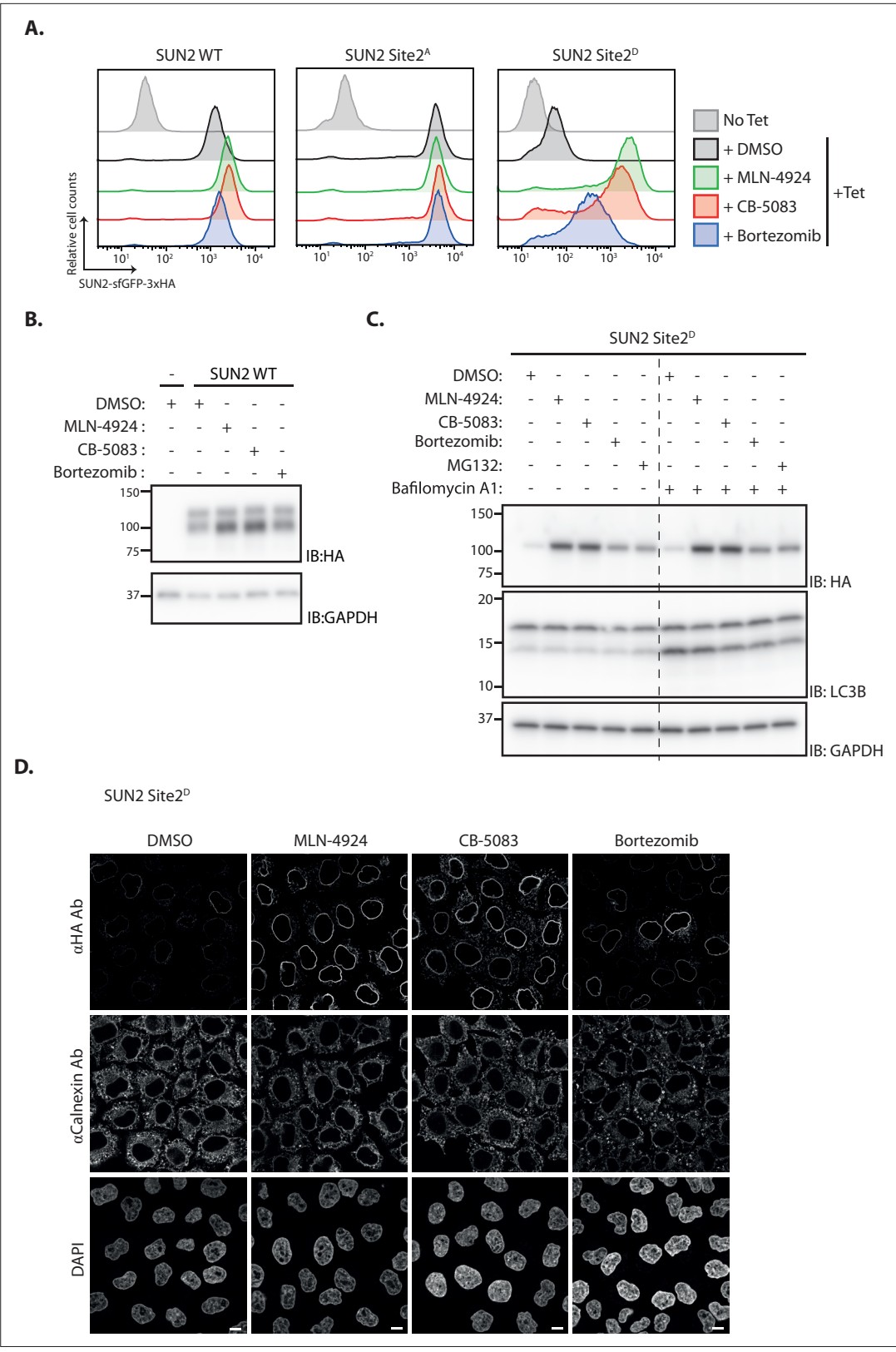

**Figure 2.** Inner nuclear membrane (INM) degradation of SUN2 by an ERAD-like process. (**A**) Flow cytometry analysis of tetracycline-induced expression of SUN2 WT, Sites 2$^A$ and 2$^D$ in HEK TF cells. Analysis was performed 24 hr post-induction in cells left untreated, incubated 4 hr with inhibitors to SCF$^{\beta TrCP}$ neddylation (MLN-4924; 1 μM), p97 (CB-5083; 2.5 μM, CB), or the proteasome (Bortezomib; 1 μM, Btz). (**B**) Analysis of SUN2 WT steady-state

*Figure 2 continued on next page*

*Figure 2 continued*

levels in HEK TF cells. Analysis was performed 24 hr post-induction in cells left untreated, incubated 4 hr with inhibitors to SCF$^{\beta TrCP}$ neddylation (MLN-4924; 1 μM), p97 (CB-5083; 2.5 μM, CB), or the proteasome (Bortezomib; 1 μM, Btz). Cell extracts were analyzed by sodium dodecyl sulfate–polyacrylamide gel electrophoresis (SDS–PAGE) and immunoblotting. SUN2 WT was detected with anti-HA antibodies. GAPDH was used as a loading control and detected with an anti-GAPDH antibody. (**C**) Analysis of SUN2 Site 2$^D$ steady-state levels in HEK TF cells. Analysis was performed 24 hr post-induction in cells incubated 4 hr with Dimethyl Sulfoxide (DMSO) (vehicle) or the inhibitors to SCF$^{\beta TrCP}$ neddylation (MLN-4924; 1 μM), p97 (CB-5083; 2.5 μM, CB), the proteasome (Bortezomib; 1 μM, Btz and 10 μM MG132), or incubated 6 hr with bafilomycin A (1 μM, Baf) that inhibits lysosomal delivery. Cell extracts were analyzed by SDS–PAGE and immunoblotting. SUN2 Site 2$^D$ was detected with anti-HA antibodies. LC3B was analyzed to confirm effectiveness of bafilomycin A treatment and was detected with an anti-LC3B antibody. GAPDH was used as a loading control and detected with an anti-GAPDH antibody. (**D**) Immunofluorescence in HeLa cells expressing SUN2 Site 2$^D$. Expression of SUN2 Site 2$^D$ was induced with Dox for 24 hr and incubated for 4 hr with DMSO (vehicle) or the inhibitors to SCF$^{\beta TrCP}$ neddylation (MLN-4924; 1 μM), p97 (CB-5083; 2.5 μM, CB), and the proteasome (Bortezomib; 1 μM, Btz). SUN2 Site 2$^D$ was detected with anti-HA antibodies. The ER marker Calnexin was detected with an anti-Calnexin antibody and DNA was labelled with 4′,6-diamidino-2-phenylindole (DAPI). Scale bar: 10 μm.

The online version of this article includes the following source data and figure supplement(s) for figure 2:

**Source data 1.** File contains original immunoblots for *Figure 2B*.

**Source data 2.** File contains original immunoblots for *Figure 2C*.

**Figure supplement 1.** Depletion of β-TrCP1/2 stabilizes SUN2 Site 2$^D$.

**Figure supplement 1—source data 1.** File contains original immunoblots for *Figure 2—figure supplement 1A*.

were generated using the MAGeCK algorithm (*Li et al., 2014*). Consistent with our earlier results, the F-Box protein βTrCP2 (encoded by FBXW11) and other core and regulatory subunits of the SCF$^{\beta TrCP}$ ubiquitin ligase, the p97 ATPase complex and the proteasome were among the highest scoring hits in our screen (*Figure 4B*). Independent sgRNAs were used to further validate some of these hits (*Figure 4—figure supplement 1A, B*) and the results demonstrated the suitability of our screen in identifying modulators of SUN2 degradation.

Another top hit from the screen was the Casein Kinase 2 β subunit (CSNK2B). As observed for other essential genes, the CSNK2B score was particularly high at day 9 and dropped at later timepoints (*Figure 4B* and *Figure 4—figure supplement 1A*). One of the genes encoding for Casein Kinase 2 α subunit (CSNK2A2) was also a hit in the screen although with a much lower score (*Figure 4—figure supplement 1A*). This is consistent with Casein Kinase 2 (CK2) functioning as a dimer of α-β subunit dimers, with CSNK2B being the sole gene encoding for the β subunit, while the α subunit is encoded by three largely redundant isoforms (*Venerando et al., 2014*). Importantly, these data raise the possibility that CK2 promotes SUN2 phosphorylation thereby facilitating its recognition and ubiquitination by SCF$^{\beta TrCP}$ ubiquitin ligase. Consistent with this model, both CK2 and SCF$^{\beta TrCP}$ ubiquitin ligase were show to have nuclear localization thereby being able to access SUN2 at the INM.

The role of CSNK2B in SUN2 degradation was validated using additional sgRNAs (*Figure 4—figure supplement 1C*). Given that the depletion of essential genes, like CSNK2B, is complicated and can give confounding effects, we used a well-characterized CK2 inhibitor, tetrabromocinnamic acid (TBCA), to study the role of CK2 in SUN2 degradation. Consistent with the gene depletion experiments, acute CK2 inhibition with TBCA resulted in an increase of the steady levels of both SUN2 WT and Site 2$^D$ mutant (*Figure 4C, D*). SUN2 WT steady levels were comparable between cells subjected to CK2 or p97 inhibition (*Figure 4C, D*). Importantly, phosphomimetic mutation in SUN2 Site 2$^D$ showed an intermediate response, while the double mutant SUN2 Sites 1$^D$,2$^D$ was largely refractory to CK2 inhibition (*Figure 4C, D*). Thus, CK2 inhibition slows down SUN2 degradation.

Next, we used immunoprecipitation to test whether CK2 regulates SUN2 degradation by controlling its interaction with the ubiquitin ligase SCF$^{\beta TrCP}$. Indeed, acute CK2 inhibition with TBCA decreased the interaction between SUN2 and βTrCP (*Figure 4E*). Together, these data indicate that CK2 activity promotes SCF$^{\beta TrCP}$ binding to SUN2 thereby facilitating its ubiquitination and degradation.

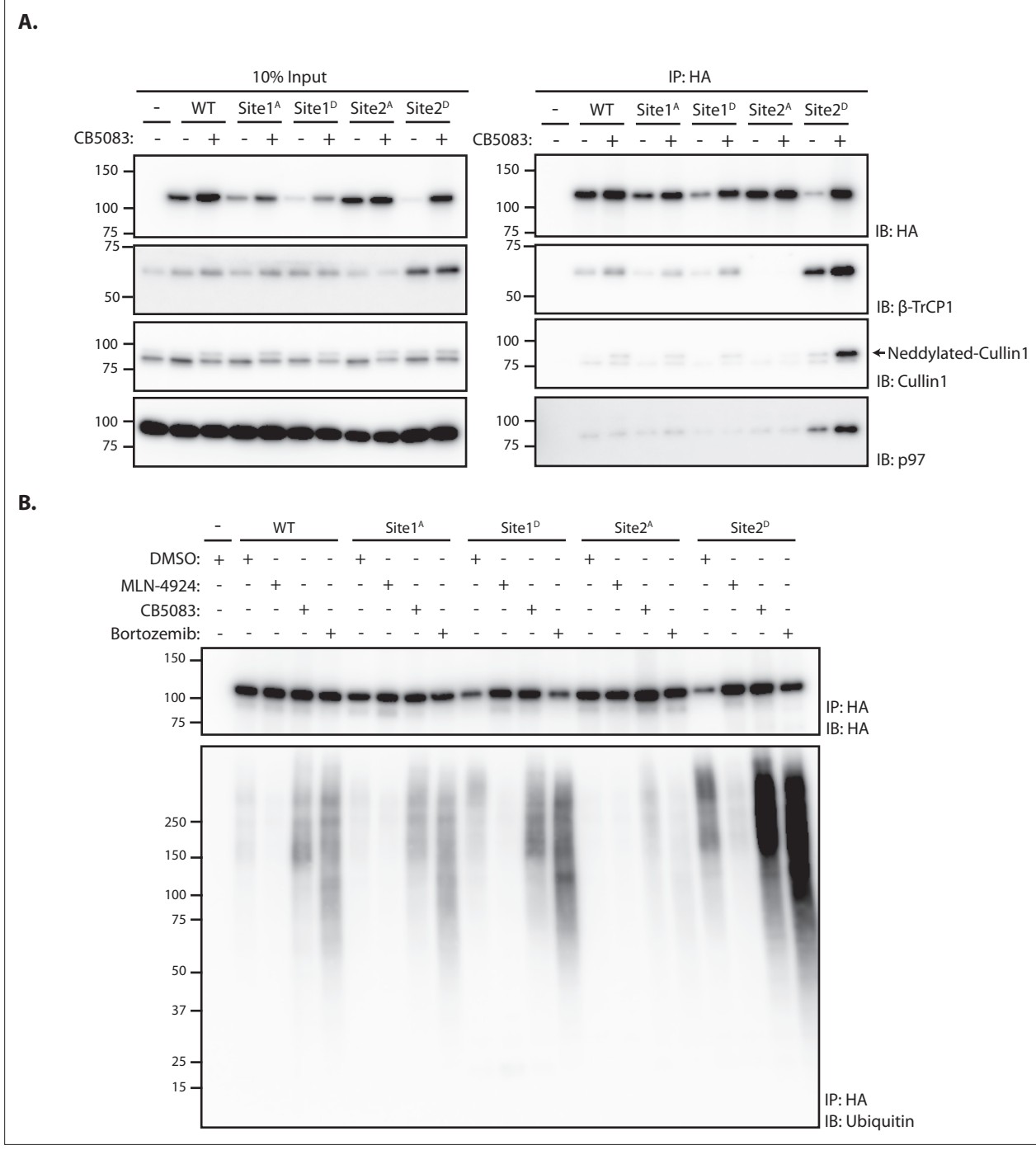

**Figure 3.** Site 2 is critical for SUN2 recognition and ubiquitination by SCF$^{\beta TrCP}$. (**A**) Binding of WT and the indicated SUN2 mutants to subunits of the ubiquitin ligase SCF$^{\beta TrCP}$ and the ATPase p97 analyzed by immunoprecipitation. Expression of SUN2 derivatives was induced with Dox for 24 hr and treated with DMSO (−) or CB-5083 (+) for 4 hr before cell lysis and immunoprecipitation with anti-HA coated beads. Eluted proteins were separated by sodium dodecyl sulfate–polyacrylamide gel electrophoresis (SDS–PAGE) and analyzed by western blotting with the indicated antibodies. (**B**) Ubiquitination of WT and the indicated SUN2 mutants in cells incubated 4 hr with DMSO (vehicle) or the inhibitors to SCF$^{\beta TrCP}$ neddylation (MLN-4924; 1 μM), p97 (CB-5083; 2.5 μM, CB), and the proteasome (Bortezomib; 1 μM, Btz). Upon immunoprecipitation with anti-HA coated beads, eluted proteins were separated by SDS–PAGE and analyzed by western blotting with anti-HA and anti-ubiquitin antibodies.

The online version of this article includes the following source data for figure 3:

**Source data 1.** File contains original immunoblots for *Figure 3A*.

**Source data 2.** File contains original immunoblots for *Figure 3B*.

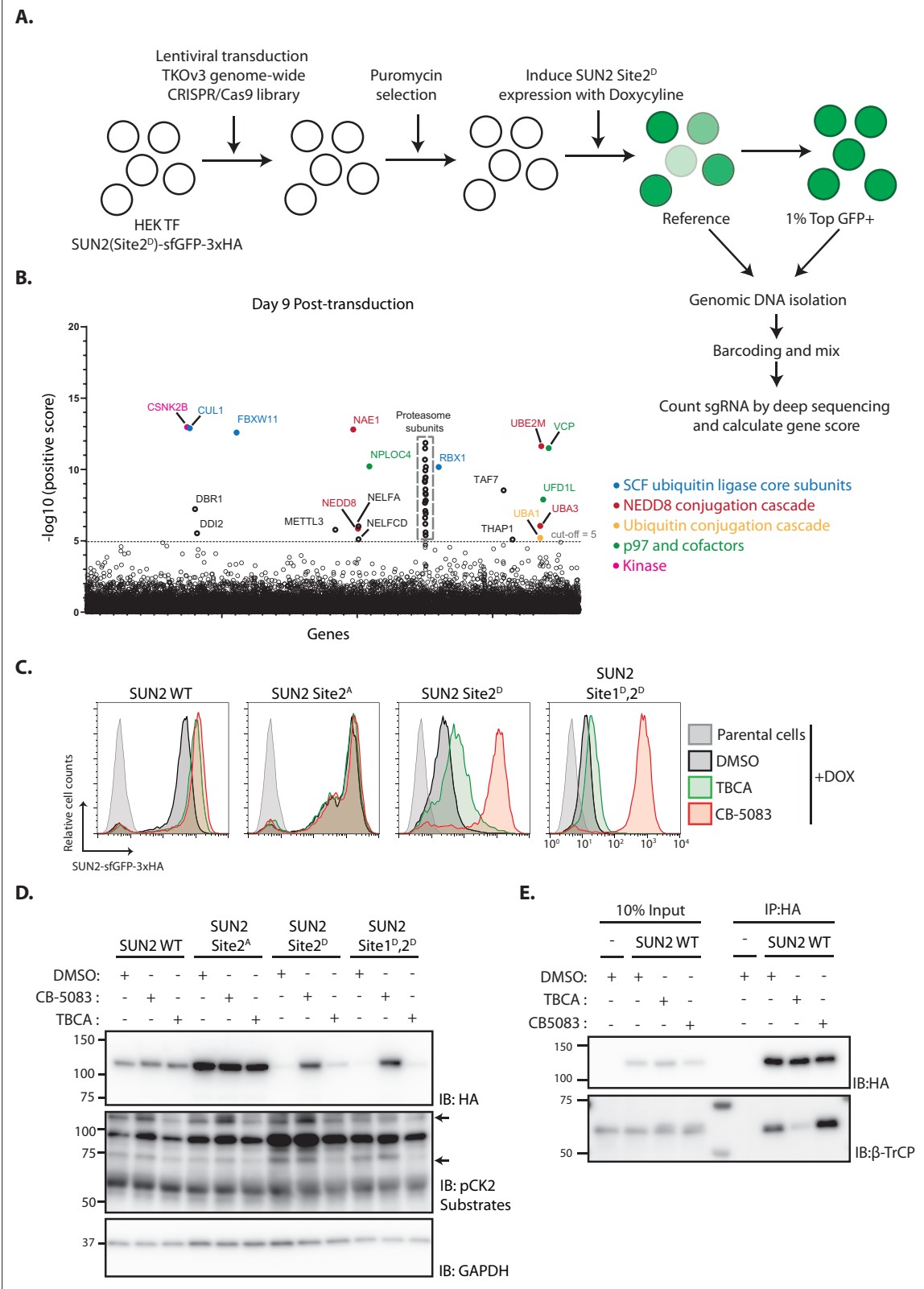

**Figure 4.** Genome-wide screening identifies Casein Kinase 2 as a positive regulator of SUN2 degradation. (**A**) Workflow of the CRISPR-Cas9 genome-wide screen. (**B**) Significance score of the genes analyzed in the screen calculated by the MAGeCK algorithm. The x-axis represents the genes in alphabetical order. The y-axis shows the −log($\alpha$RRA) significance value. The −log($\alpha$RRA) cut-off was arbitrarily set at 5 (dashed line). Significantly enriched genes are annotated and colour coded: genes related to SCF ubiquitin ligase core subunits (blue), NEDD8 conjugation cascade (red), ubiquitin

*Figure 4 continued on next page*

*Figure 4 continued*

conjugation cascade (orange), p97 and co-factors (green), and kinase (pink). Proteasome subunits are shown in a grey dotted box. Flow cytometry (**C**) and western blotting (**D**) analysis of Dox-induced expression of WT, Sites 2$^A$, 2$^D$, and Sites 1$^D$, 2$^D$ SUN2 in HEK TF cells. Analysis was performed 24 hr post-induction in cells incubated 6 hr with inhibitors to Casein Kinase 2 (tetrabromocinnamic acid, TBCA; 100 µM) and p97 (CB-5083; 2.5 µM, CB). SUN2 derivatives were detected with anti-HA antibody. CK2 inhibition was confirmed by blotting with anti-phospho-CK2 substrates antibody. GAPDH was used as a loading control and detected with an anti-GAPDH antibody. (**E**) Immunoprecipitation of SUN2 WT from cells incubated 6 hr with DMSO (vehicle), Casein Kinase 2 inhibitor (TBCA; 100 µM), and p97 inhibitor (CB-5083; 2.5 µM, CB). Cell lysates were immunoprecipitated with anti-HA coated beads. Eluted proteins were separated by sodium dodecyl sulfate–polyacrylamide gel electrophoresis (SDS–PAGE) and analyzed by western blotting with the indicated antibodies.

The online version of this article includes the following source data and figure supplement(s) for figure 4:

**Source data 1.** File contains original immunoblots for *Figure 4D*.

**Source data 2.** File contains original immunoblots for *Figure 4E*.

**Figure supplement 1.** Analysis of genome-wide CRISPR/Cas9 genetic screening.

**Figure supplement 1—source data 1.** File contains original immunoblots for *Figure 4—figure supplement 1*.

## The phosphatase CTDNEP1 is a negative regulator of SUN2 degradation

Under normal conditions, SUN2 is a relatively stable protein (*Figure 5C*; *Buchwalter et al., 2019*). Therefore, we hypothesized that CK2-dependent SUN2 degradation is counteracted by the activity of a protein phosphatase. Given its localization to the INM and phosphatase activity towards Serine/Threonine residues, the CTDNEP1 was an attractive candidate to regulate SUN2 turnover (*Bahmanyar et al., 2014*; *Kim et al., 2007*). To test this possibility, we analyzed the levels of endogenous SUN2 upon CTDNEP1 depletion. Consistent with our hypothesis, the steady-state levels of endogenous SUN2 and of SUN2 WT transgene were strongly reduced in CTDNEP1 depleted cells (*Figure 5*). This effect was specific to SUN2 as the levels of other INM proteins such as SUN1 (*Figure 5A, B*) and LBR (*Figure 5—figure supplement 1A*) remained unchanged. In CTDNEP1 KO cells, the lower levels of endogenous SUN2 (*Figure 5C*) and SUN2 WT transgene (*Figure 5—figure supplement 1B*) were due to their accelerated degradation, as monitored by cycloheximide chase experiments. Importantly, increased turnover of endogenous SUN2 (*Figure 5—figure supplement 1C, D*) and transgenic SUN2 WT (*Figure 5D*) in CTDNEP1 depleted cells was reversed by inhibition of the SCF$^{βTrCP}$ ubiquitin ligase, the p97 ATPase and the proteasome. Moreover, regulation of SUN2 turnover required CTDNEP1 phosphatase activity. As shown in *Figure 5E* and *Figure 5—figure supplement 1E*, re-expression of wild-type CTDNEP1 (CTDNEP1-Flag) but not of a catalytically inactive CTDNEP1 (CTDNEP1(PD)-Flag) mutant in CTDNEP1 KO cells restored the levels of both endogenous and transgenic SUN2. Thus, the nuclear envelope phosphatase CTDNEP1 is a negative regulator of SUN2 degradation.

Together with our earlier results, these data also suggest a model in which SUN2 levels are controlled by a phospho-switch, with the kinase CK2 promoting SUN2 degradation and the phosphatase CTDNEP1 counteracting it. In support of this model, the reduction of SUN2 levels in CTDNEP1 KO cells is counteracted by acute CK2 inhibition with TBCA (*Figure 5F* and *Figure 5—figure supplement 1F*). A second prediction of the model is that the SUN2 Site 2$^A$, which cannot be phosphorylated in the main site controlling SUN2 degradation, is insensitive to the loss of CTDNEP1. Indeed, depletion of CTDNEP1 while resulting in low levels of SUN2 WT transgene, did not affect SUN2 Site 2$^A$ mutant level (*Figure 5G* and *Figure 5—figure supplement 1G*). Importantly, the levels of endogenous SUN2 were still reduced in these cells, confirming the depletion of CTDNEP1 (*Figure 5G*). Therefore, CK2 and CTDNEP1 have opposing effects on SUN2 turnover.

## Aberrant nuclear architecture and function by expression of non-degradable SUN2

Next, we investigated the importance of SUN2 regulated degradation for nuclear morphology and function. To this end, we analyzed nuclear morphology in HeLa cells expressing either SUN2 WT or the non-degradable SUN2 Site 2$^A$ mutant. In parental HeLa cells, most nuclei display a smooth or slightly wrinkled surface. Expression of SUN2 WT from a Dox-inducible promoter for 24 hr increased the fraction of wrinkled nuclei as expected (*Donahue et al., 2016*). A small fraction of SUN2 WT-expressing cells showed aberrant nuclei, which were highly convoluted and multi-lobed. Nearly half of the cells

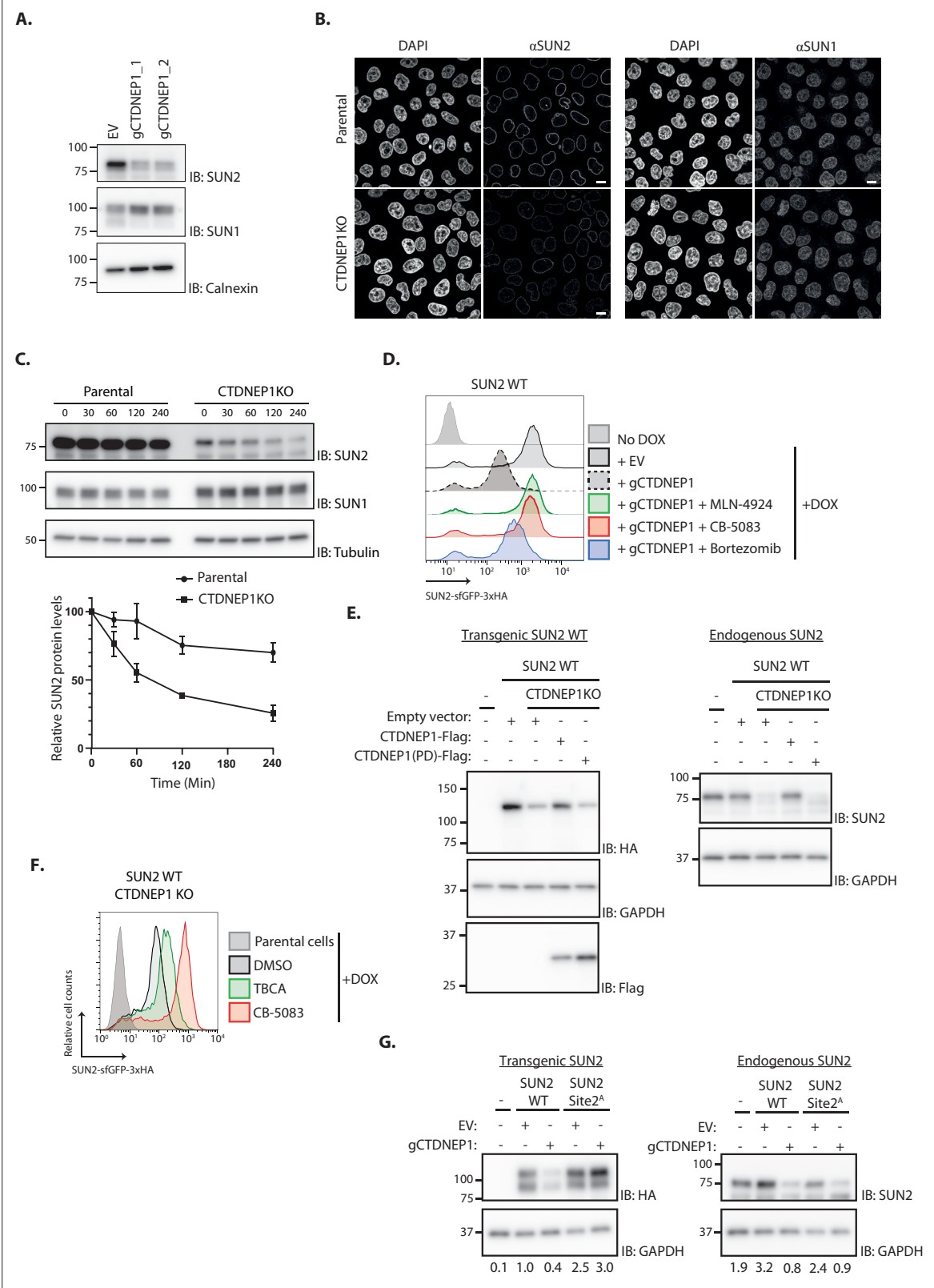

**Figure 5.** The phosphatase CTDNEP1 is a negative regulator of SUN2 degradation. (**A**) Steady-state levels of endogenous SUN2 in HEK TF transfected with an empty vector (EV) or sgRNAs targeting CTDNEP1. Cell extracts were analyzed by sodium dodecyl sulfate–polyacrylamide gel electrophoresis (SDS–PAGE) and immunoblotting with anti-SUN2 and anti-SUN1 antibodies. Calnexin was used as a loading control and detected with an anti-Calnexin antibody. (**B**) Immunofluorescence in parental and CTDNEP1 KO HeLa cells. Expression of SUN2 Site 2$^D$ was induced with Dox for 24 hr. Endogenous

*Figure 5 continued on next page*

*Figure 5 continued*

SUN2 and SUN1 were detected with anti SUN2 and anti-SUN1 antibodies, respectively. DNA was labelled with DAPI. Scale bar: 10 μm. (**C**) Analysis of endogenous SUN2 stability after inhibition of protein synthesis by cycloheximide (CHX) in parental and CTDNEP1 KO HEK TF cells. Cell extracts were analyzed by SDS–PAGE and immunoblotting. Endogenous SUN2 and SUN1 were detected with anti SUN2 and anti-SUN1 antibodies, respectively. Tubulin was used as a loading control and detected with an anti-Tubulin antibody. Three independent experiments were quantified on the right. (**D**) Flow cytometry analysis of SUN2 WT in HEK TF cells transfected with an empty vector (EV) or with a sgRNA targeting CTDNEP1. Analysis was performed 24 hr post-Dox induction in cells left untreated or incubated 4 hr with inhibitors to SCF$^{βTrCP}$ neddylation (MLN-4924; 1 μM), p97 (CB-5083; 2.5 μM, CB), or the proteasome (Bortezomib; 1 μM, Btz). (**E**) Analysis of transgenic (+Dox; left) and endogenous SUN2 (No Dox; right) steady-state levels in HEK TF SUN2 WT-expressing parental and CTDNEP1 KO cells transfected with an empty vector (EV), a vector encoding FLAG-tagged WT CTDNEP1 or phosphatase dead CTDNEP1 (PD). Exogenous SUN2 WT and endogenous SUN2 were detected with anti-HA and anti-SUN2 antibodies, respectively. CTDNEP1-FLAG was detected with anti-FLAG antibodies. GAPDH was used as a loading control and detected with an anti-GAPDH antibody. (**F**) Flow cytometry analysis of SUN2 WT in CTDNEP1 KO HEK TF cells. Analysis was performed 24 hr post-Dox induction in cells incubated 6 hr with DMSO (vehicle), Casein Kinase 2 inhibitor (tetrabromocinnamic acid, TBCA; 100 μM), and p97 inhibitor (CB-5083; 2.5 μM, CB). (**G**) Analysis of transgenic (+Dox; left) and endogenous SUN2 (No Dox; right) steady-state levels in HEK TF SUN2 WT or Site 2 $^A$ cells transfected with an empty vector (EV) or with a sgRNA targeting CTDNEP1. Transgenic and endogenous SUN2 were detected with anti-HA and anti-SUN2 antibodies, respectively. GAPDH was used as a loading control and detected with an anti-GAPDH antibody. The abundance of SUN2 in relation to GAPDH was quantified and is shown below each lane.

The online version of this article includes the following source data and figure supplement(s) for figure 5:

**Source data 1.** File contains original immunoblots for *Figure 5A*.

**Source data 2.** File contains original immunoblots for *Figure 5C*.

**Source data 3.** File contains original immunoblots for *Figure 5E*.

**Source data 4.** File contains original immunoblots for *Figure 5G*.

**Figure supplement 1.** Characterization of the CTDNEP1 role in SUN2 degradation.

**Figure supplement 1—source data 1.** File contains original immunoblots for *Figure 5—figure supplement 1B*.

**Figure supplement 1—source data 2.** File contains original immunoblots for *Figure 5—figure supplement 1C*.

**Figure supplement 1—source data 3.** File contains original immunoblots for *Figure 5—figure supplement 1D*.

**Figure supplement 1—source data 4.** File contains immunoblots for *Figure 5—figure supplement 1F*.

expressing non-degradable SUN2 Site 2$^A$ displayed these highly aberrant nuclei, which were never observed in parental HeLa cells (*Figure 6B*). Importantly, nuclear morphology greatly improved upon shutting off expression of SUN2 WT for 24 hr. In contrast, aberrant nuclear morphologies persisted even 24 hr after shutting-off SUN2 Site 2$^A$ expression, consistent with the long half-life of this protein (*Figure 6A, B* and *Figure 6—figure supplement 1A*). In the aberrant nuclei, SUN2 Site 2$^A$ was distributed heterogeneously, accumulating in patches or lines. Besides SUN2 Site 2$^A$, these structures were enriched in certain INM proteins, such as LBR and EMD. However, other INM proteins like LAP 2β and SUN1 were unaffected (*Figure 6C*). Thus, accumulation of non-degradable SUN2 drives aberrant NE architecture and redistribution of specific INM proteins.

SUN2 was implicated in facilitating NE disassembly during mitosis (*Turgay et al., 2014*). We wondered if the changes in nuclear architecture due to accumulation of non-degradable SUN2 Site 2$^A$ would impact mitotic progression. Timelapse fluorescence microscopy showed that cells expressing SUN2 WT and Site 2$^A$ progress through the various stages of mitosis with normal kinetics (*Figure 6— figure supplement 1B*). In particular, NE breakdown, chromosome alignment at the metaphase plate and anaphase onset occurred with similar timings. In addition, the frequency of multipolar spindles was low in both cases (*Figure 6—figure supplement 1B*). However, cells expressing SUN2 Site 2$^A$ showed higher frequency of lagging chromosomes during anaphase (*Figure 6—figure supplement 1C, D*). Curiously, these cells also showed an increased association of chromosomes with the nuclear membrane (*Figure 6—figure supplement 1D*). Thus, accumulation of non-degradable SUN2 may impact chromosome dynamics, potentially by interfering with interactions between chromosomes and the INM.

SUN2 has also been implicated in DNA damage repair, particularly in the repair of double strand breaks (*Lei et al., 2012*; *Lottersberger et al., 2015*). While the precise role of SUN2 in DNA repair is ill defined, it may control the mobility of damaged loci thereby affecting the efficiency of the repair process (*Lottersberger et al., 2015*). Therefore, we asked whether accumulation of non-degradable SUN2 Site 2$^A$ impacts the cell's ability to repair DNA double strand breaks. To assess the efficiency of

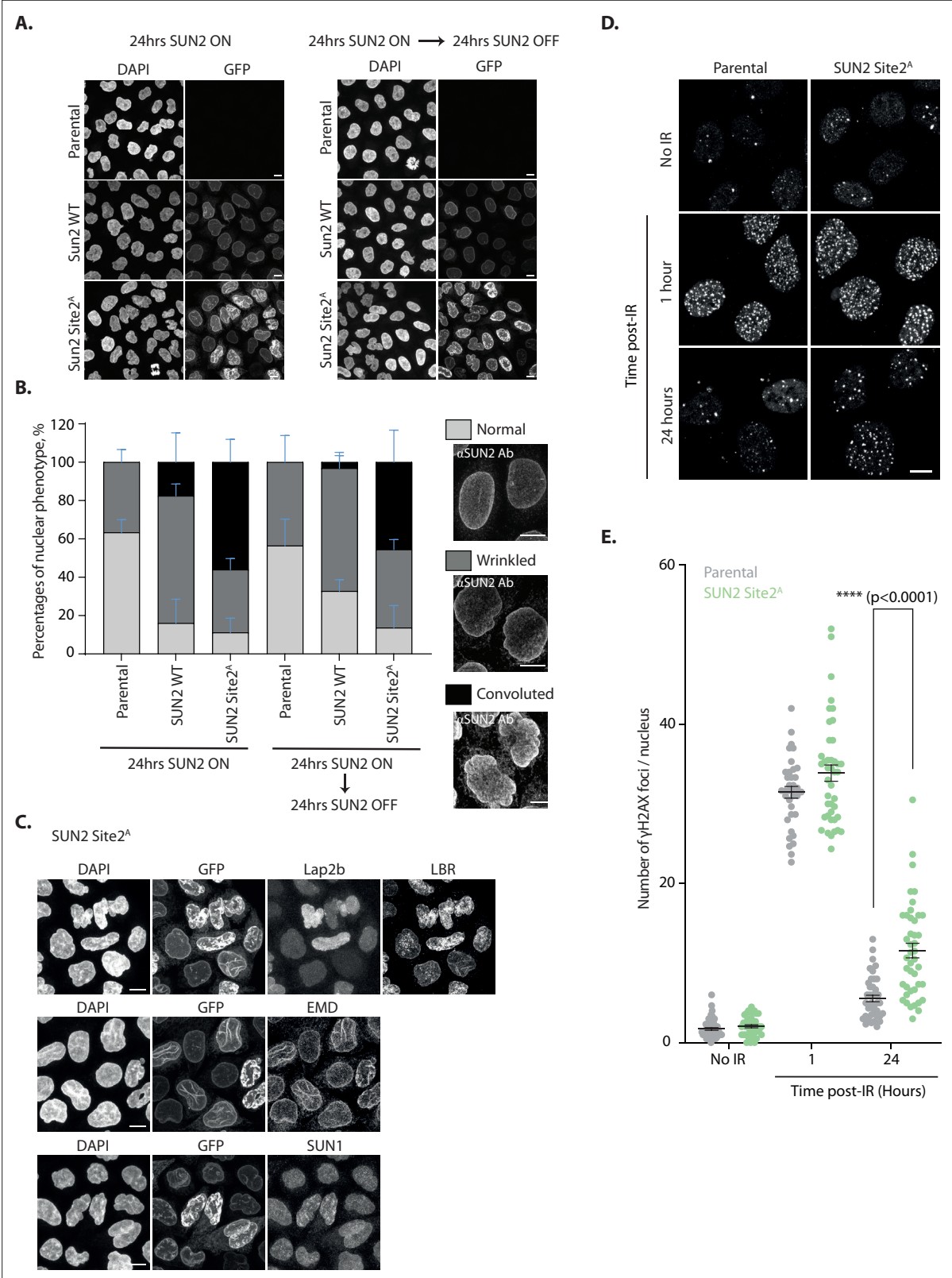

**Figure 6.** Accumulation of non-degradable SUN2 results in aberrant nuclear architecture and function. (**A**) Micrographs of parental HeLa or cells expressing SUN2 WT or Site 2^A. Cells were analyzed 24 hr after inducing SUN2 expression (left) or 24 hr after turning off SUN2 expression (right). SUN2 was detected with green fluorescent protein (GFP) and DNA was labelled with DAPI. (**B**) Quantification of nuclear morphology in HeLa cells as in (**A**). Three independent experiments, with 50–150 cells for each condition in each repeat were quantified. Representative examples of nuclear morphologies

*Figure 6 continued on next page*

*Figure 6 continued*

are shown on the right. Error bars indicate standard deviation. (**C**) Immunofluorescence in HeLa cells upon induction of SUN2 Site 2$^A$ expression for 24 hr. SUN2 Site 2$^A$ was detected with GFP. Anti-LAP2β, -LBR, -EMD, and -SUN1 antibodies were used to detect these inner nuclear membrane (INM) proteins. DNA was labelled with DAPI. (**D**) Immunofluorescence in U2OS parental and SUN2 Site 2$^A$-expressing cells irradiated with 5 Gy ionizing radiation (IR). Time after irradiation is indicated. Non-irradiated cells were used as control (shown on the top). SUN2 Site 2$^A$ expression was induced 24 hr prior irradiation. γH2AX foci were detected with an anti-γH2AX antibody. Scale bar: 10 μm. (**E**) Quantification of γH2AX foci per nucleus detected in (**D**).Three independent experiments were analyzed and 30–40 nuclei were quantified for each condition in each replicate. Error bar represents standard error of the means (SEM). Ordinary one-way analysis of variance (ANOVA) and Tukey's multiple comparisons were performed (****$p < 0.0001$).

The online version of this article includes the following figure supplement(s) for figure 6:

**Figure supplement 1.** Analysis of the accumulation of non-degradable SUN2 in mitotic progression.

DNA repair we monitored the DNA damage marker γH2AX (Ser139-phosphorylated histone H2AX) (*Rogakou et al., 1998*) in U2OS cells subjected to ionizing radiation (IR) to trigger double strand breaks. Parental and SUN2 Site 2$^A$-expressing cells have comparable levels of γH2AX both prior and soon after IR treatment (*Figure 6D, E*). This indicates that expression of non-degradable SUN2 is (1) insufficient to trigger DNA damage; and (2) does not prevent cells from mounting a DNA damage response upon induction of double strand breaks with IR. At 24 hr post IR, parental cells show nearly basal levels of γH2AX indicating that DNA damage has been repaired. In contrast, the levels of γH2AX in SUN2 Site 2$^A$ are higher 24 hr post IR, likely due to unresolved DNA damage. Thus, accumulation of non-degradable SUN2 Site 2$^A$ appears to compromise efficient DNA repair.

Altogether, these data indicate that accumulation of non-degradable SUN2 affects nuclear architecture and impacts nuclear functions.

## Discussion

We uncovered that the INM protein SUN2, a subunit of the LINC complex, undergoes regulated turnover (*Figure 7*). The kinase CK2 and the phosphatase CTDNEP1 act, respectively, as positive and negative regulators of SUN2 degradation by influencing the phosphorylation state of non-canonical binding sites for the SCF$^{βTrCP}$ ubiquitin ligase, primarily Site 2. Binding of SCF$^{βTrCP}$ to phosphorylated Site 2 results in SUN2 ubiquitination and the recruitment of the p97 ATPase complex, which facilitates SUN2 membrane extraction and delivery to the proteasome for degradation. Cells expressing non-degradable SUN2 display defects in nuclear architecture highlighting the importance of this ERAD-like process in maintaining INM protein homeostasis in mammalian cells.

While SUN2 and the LINC complex play important functions in cell homeostasis, conditions exist in which the forces they transduce into the nucleus are deleterious. It has been shown that aberrant DNA

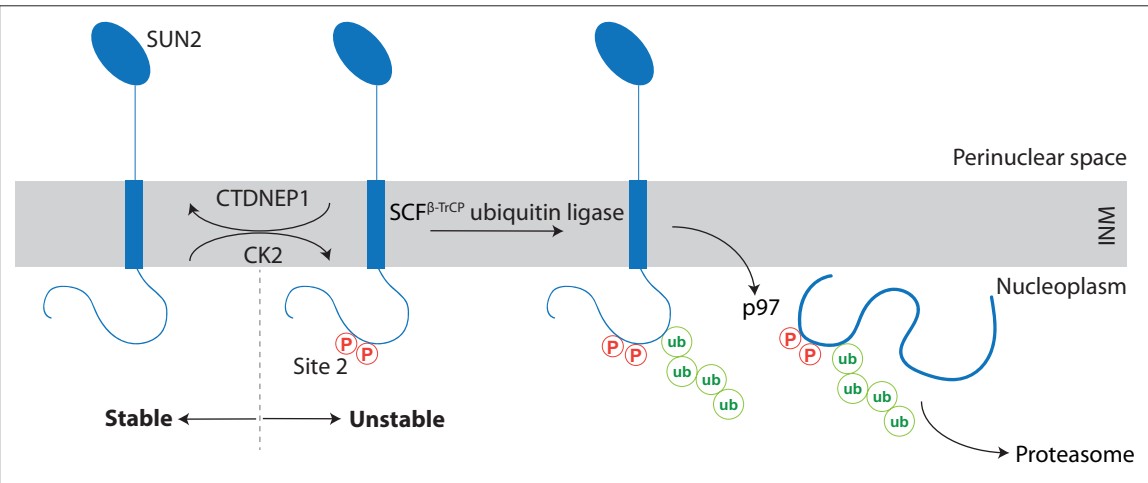

**Figure 7.** A kinase-phosphatase balance regulates ER-associated degradation of SUN2 from the inner nuclear membrane (INM). The scheme illustrates the opposing effects of the kinase CK2 and the phosphatase CTDNEP1 on SUN2 degradation. Phosphorylation of SUN2 Site 2 promotes the binding of SCF$^{βTrCP}$ ubiquitin ligase and subsequent SUN2 ubiquitination, membrane extraction by the p97 ATPase complex and delivery to the proteasome for degradation. SUN2 is depicted as a monomer for simplicity however it functions as a homotrimer.

repair resulting from increased chromatin mobility at sites of double strand breaks depends on the LINC complex (*Lottersberger et al., 2015*). Consistent with these data, we observe that expression of non-degradable SUN2 results in impaired repair of double strand breaks. Downregulation of the LINC complex resulting in the decoupling of the nucleus from the cytoskeleton has been observed in cells under extreme cycles of mechanical load (*Gilbert et al., 2019*). This nuclear decoupling protects DNA from mechanical-induced damage and appears to involve a decrease in SUN2 levels by some unknown mechanism. Thus, it is appealing to consider that downregulation of LINC complexes by localized SUN2 degradation at sites of DNA damage or extreme mechanical load may facilitate DNA repair or prevent mechanical-induced damage. Future studies should test these possibilities directly.

Mechanistically, the involvement of a soluble ubiquitin ligase, SCF$^{βTrCP}$, in SUN2 degradation presents an important difference to conventional ERAD, which is normally mediated by transmembrane ubiquitin ligases (*Christianson and Carvalho, 2022*). However, the participation of SCF$^{βTrCP}$ in an ERAD-like process is not unprecedented. In HIV infected cells, SCF$^{βTrCP}$ promotes the degradation of the integral membrane protein CD4 (*Florence et al., 1998*). In this case, virally encoded Vpu protein functions as an adaptor, binding to the substrate CD4 in the ER membrane and recruiting the ubiquitin ligase SCF$^{βTrCP}$ via the cytosolic tail (*Magadán et al., 2010*; *Margottin et al., 1998*; *Zhang et al., 2013*). Based on these observations, it will be interesting to test whether SCF$^{βTrCP}$ has a more widespread role in the degradation of membrane proteins. Interestingly, a soluble ubiquitin ligase is also implicated in downregulating the yeast SUN domain-containing protein Mps3 by an ERAD-like process (*Koch et al., 2019*). In this case, the degradation of Mps3, which is important for the distribution of nuclear pore complexes and spindle pole bodies (the yeast equivalent to centrosomes) within the INM (*Chen et al., 2014*; *Jaspersen et al., 2006*), facilitates cell cycle progression. Like in the case of SUN2, accumulation of non-degradable Mps3 affects nuclear morphology (*Koch et al., 2019*).

Extraction of ERAD substrates from the membrane into the cytosol, a step known as retrotranslocation, is driven by the pulling force of p97 ATPase and facilitated by transmembrane segments of ERAD ubiquitin ligases or accessory factors (*Christianson and Carvalho, 2022*; *Wu and Rapoport, 2018*). In the case of a soluble ubiquitin ligase like SCF$^{βTrCP}$, how membrane substrates are retrotranslocated is unclear. Our genome-wide CRISPR screen did not identify any membrane proteins that could potentially mediate SUN2 retrotranslocation, suggesting redundancy between multiple components. Alternatively, the pulling force imposed by p97 may be sufficient to extract SUN2 from the INM membrane.

Regulated protein degradation by ERAD is well documented, particularly in the control of sterol biosynthesis (*Christianson and Carvalho, 2022*; *Johnson and DeBose-Boyd, 2018*). In these cases, the degradation of rate limiting enzymes depends on the concentration of specific sterol metabolites in the ER membrane and functions as a feedback mechanism controlling sterol biosynthesis. To our knowledge, SUN2 is the first substrate shown to be controlled by a kinase/phosphatase balance. Given the requirement of phosphorylation for SCF$^{βTrCP}$ binding to substrates, the involvement of a kinase was expected (*Frescas and Pagano, 2008*). In fact, CK2 has been implicated in the degradation of other SCF$^{βTrCP}$ substrates such as Vpu-mediated degradation of CD4 (*Florence et al., 1998*) and Cyclin F (*Mavrommati et al., 2018*). In contrast, the involvement of CTDNEP1 is unexpected as this phosphatase has been mainly implicated in dephosphorylation of Lipin, a key regulator of lipid metabolism (*Barger et al., 2022*; *Zhang and Reue, 2017*). In particular, CTDNEP1 was recently implicated in coordinating membrane biogenesis with cell division and loss of CTDNEP1 results in excess membrane production which affect chromosome dynamics (*Merta et al., 2021*). A role for SUN2 in lipid metabolism was also recently proposed (*Lee et al., 2022*), suggesting that the convoluted nuclei observed in cells expressing non-degradable SUN2 may result from excessive production of NE membranes. Thus, future studies should explore further how the function of CTDNEP1 in controlling SUN2 levels relates to its role in lipid regulation. Another important open question is whether CTDNEP1 and CK2 act directly on SUN2.

While studies in rats and cultured cells suggest that INM proteins are generally long lived (*Buchwalter et al., 2019*; *Toyama et al., 2013*), our results show that the INM proteome can be remodelled in a highly regulated fashion. Other recent examples show that certain conditions, like mutations (*Tsai et al., 2016*) or ER stress (*Buchwalter et al., 2019*), can trigger the turnover of specific INM proteins. Thus, it appears that the turnover of INM proteins is likely to be temporal and spatially restricted, allowing fine-tuning of INM properties without jeopardizing its overall structure and critical functions in nuclear organization.

# Methods

## Plasmids

For stable integration of SUN2 constructs into the FRT sites of Flp-In T-Rex HEK cells, SUN2 cDNAs were cloned into the pcDNA5-FRT-TO plasmid (Invitrogen). For CTDNEP1-rescue experiments a lentiviral vector was used with the expression of CTDNEP1 being driven by the EF-1α promoter. For experiments in HeLa and U2OS cells, we used a lentiviral vector with an inducible doxycycline-responsive promoter driving the expression of SUN2 WT and variants. For gene knockouts, a lentiviral pSicoR plasmid with U6 promoter driving sgRNA expression and EF-1a promoter driving the expression of Puromycin-T2A-Flag-Cas9 was used.

## Cell culture

All cells were grown at 37°C, 5% $CO_2$ in DMEM medium (Sigma-Aldrich) supplemented with L-glutamine (2 mM; Gibco), penicillin–streptomycin (10 units/ml; Gibco), and 10% Fetal Calf Serum (FCS) (Gibco). Cells were regularly tested for mycoplasma contamination.

## Generation of Flp-In T-Rex HEK (HEK TF) cell lines

Cell lines were made as per the manufacturer's protocol.

## Lentiviral production and transduction

Lenti-X cells were co-transfected with lentiviral and packaging plasmids using the Mirus LT1 transfection reagent in 24-well tissue culture plate format. Seventy-two hours later, the media was harvested. For transduction, cells were seeded in a 24-well tissue culture plate ($1.5 \times 10^5$ cells for HEK and $7.5 \times 10^4$ cells for HeLa). The next day, 200 µl of lentivirus was added onto the cells. Twenty-four hours later, the media was replaced and cells were grown for further 24 hr. Cells were expanded to 10 cm tissue culture-treated petri dish in the presence of antibiotic selection (zeocin or blastocidin for either the constitutive or the inducible lentiviral plasmids, respectively).

## Co-immunoprecipitation

Expression of SUN2 constructs in HEK293 TF cells was induced with 1 µg/ml doxycycline 24 hr prior to lysis. Cells were grown to 80% confluency in 6-well format, washed in TBS (50 mM Tris–HCl pH 7.5, 150 mM NaCl) once, and lysed in 1% Decyl Maltose Neopentyl Glycol (DMNG) (Anatrace) lysis buffer (50 mM Tris–HCl pH 7.5, 150 mM NaCl) containing cOmplete protease inhibitor cocktail (Roche) and 5 mM $N$-ethylmaleimide (NEM). Cell suspension was solubilized on a rotating wheel for 2 hr at 4°C. Cell debris and nuclei were pelleted at $20,000 \times g$ for 20 min at 4°C. Postnuclear supernatants were incubated for 2 hr with pre-equilibrated 20 µl anti-HA antibody magnetic beads slurry (Sigma-Aldrich). After three 10 min washes in 0.2% DMNG washing buffer (50 mM Tris–HCl pH 7.5, 150 mM NaCl) on rotating wheel at 4°C, proteins were eluted from the beads in 1× sample buffer for 15 min at 65°C and collected into fresh tubes using magnetic racks. Eluates were subsequently supplemented with 100 mM Dithiotheritol (DTT).

## Substrate ubiquitination assay

Expression of SUN2 constructs in HEK293 TF cells was induced with 1 µg/ml Doxycycline 24 hr prior to lysis. Cells at around 80% confluency in a 10-cm dish were lysed in RadioImmunoPrecipitation Assay (RIPA) lysis buffer (50 mM Tris–HCl pH 7.5, 150 mM NaCl, 1% Triton X-100, 0.5% sodium deoxycholate, 0.1% sodium dodecyl sulfate [SDS]) containing NEM (5 mM) and cOmplete protease inhibitor cocktail (Roche). Cell suspension was solubilized on a rotating wheel for 2 hr at 4°C. Cell debris and nuclei were pelleted at $20,000 \times g$ for 20 min at 4°C. Postnuclear supernatants were incubated for 2 hr with pre-equilibrated 30 µl anti-HA antibody magnetic beads slurry (Sigma-Aldrich). After three 15 min washes in RIPA buffer on a rotating wheel at 4°C, proteins were eluted from the beads in 1× sample buffer for 15 min at 65°C and collected into fresh tubes using magnetic racks. Eluates were subsequently supplemented with 100 mM DTT.

## Translation shut-off experiments

Expression of SUN2 constructs in Flp-In TRex HEK cells was induced with 1 µg/ml doxycycline 24 hr prior to translation shut-off experiments. Cells were incubated with cycloheximide (50 µg/ml) for the

indicated timepoints, after which cells were lysed in 1× sample buffer containing Benzonase (Sigma-Aldrich), cOmplete protease inhibitor cocktail (Roche), and DTT. Lysates were incubated for 30 min at 37°C, after which proteins were separated by SDS–polyacrylamide gel electrophoresis (PAGE). Immunoblotting was performed as described below. Representative images of at least three independent experiments are shown. Error bars represent the standard deviation.

## Immunoblotting

Samples were incubated at 65°C for 10 min, separated by SDS–PAGE (Bio-Rad) and proteins were transferred to PVDF membranes (Bio-Rad). Membranes were blocked in 5% Milk or bovine serum albumin (BSA) in phosphate-buffered saline (PBS)-Tween20 buffer and then probed with primary antibodies overnight at 4°C on shaker. Secondary antibodies were performed at RT for 1 hr either in 0.5% Milk or BSA in PBS-Tween20 buffer. Membranes were developed by ECL (Western Lightning ECL Pro, PerkinElmer), and visualized using an Amersham Imager 600 (GE Healthcare Life Sciences).

## Immunofluorescence

HeLa cells are seeded ($7.5 \times 10^4$ cells) in DMEM supplemented with Pen/Strep onto round coverslips (13 mm, #1.5) in a 12-well tissue culture-treated plate. Twenty-four hours later, media was replaced to 1 µg/ml doxycycline-containing DMEM supplemented with Pen/Strep and incubated for further 24 hr. For Dox wash out experiments, $5.0 \times 10^4$ cells were seeded to allow for an additional day for doxycycline washout protocol. Media was aspirated off and washed once with PBS. Cells were fixed with 4% methanol-free Paraformaldehyde (PFA) for 10 min at RT, followed by three PBS washes. Fixed cells were then treated with 10 mM glycine for 5 min, followed by three PBS washes. Fixed cells were incubated in blocking buffer (BB; 1% BSA, 0.1% Saponin in PBS) for 30 min. Primary antibodies cocktails were prepared in BB. For primary antibody staining, coverslip was carefully placed down onto pre-spotted antibodies cocktail (20 µl/cover slip) on clean parafilm and incubated for 1 hr underneath home-made aluminum foil-covered moisturized chamber. Cover slips were washed three times with BB and secondary antibody staining was performed as was done for primary staining. Cover slips, then were incubated in DAPI-containing PBS for 5 min and washed three times with PBS. Coverslips were mounted onto glass slide in non-hardening mounting media and sealed with nail polish.

## Confocal fluorescence microscopy

Fixed cells on slides after immunofluorescence were imaged at 21°C using an inverted Zeiss 880 microscope fitted with an Airyscan detector using ZEN black software. The system was equipped with Plan-Apochromat ×63/1.4-NA oil lens, with an immersion oil (Immersol W 2010, Carl Zeiss; refractive index of 1.518). 488 nm argon and 405, 561, and 633 nm solid-state diode lasers were used to excite fluorophores. Z-sections with 0.37-µm-thick intervals were collected. The oil objective was covered with an immersion oil (ImmersolT 518F, Carl Zeiss) with a refractive index of 1.518.

Microscopy images with CZI file format were analyzed using ImageJ (bundled with Java 1.8.0_172) software. Scoring of nuclear morphology was done after maximum intensity projection image processing.

## Live cell imaging

Timelapse imaging of paired GFP-SUN2 WT and Site 2 [A] cells was performed on a DeltaVision Elite light microscopy system. Fluorescence images were collected on a 512 × 512-pixel electron-multiplying charge-coupled device camera (QuantEM; Photometrics) using the software package softWoRx (GE Healthcare). Cells were seeded on two-chambered glass-bottom dishes (Lab-Tek) at 30,000 per well. Cell media was replaced with FluoroBrite DMEM media supplemented with 10% fetal bovine serum (FBS), GlutaMAX and SiR-Hoechst at a final concentration of 100 nM 8 hr prior to imaging. For imaging, cells were placed in a 37°C and 5% $CO_2$ environmental chamber (Tokai Hit). Per field of view, seven planes were captured 2 µm apart every 5 min, with light powers at 5% and 15 ms exposures. Maximum-intensity projections were performed using softWoRx, with image cropping and analysis performed using Fiji. Three independent experiments were conducted, with more than 45 cells per cell line being analyzed in each repeat. Statistical analysis and graphing were performed on GraphPad Prism v9.0. Initiation of prophase, NE breakdown, metaphase (alignment) and anaphase were analyzed as described (*Hayward et al., 2019*). Cells with delayed chromosome release from

nuclear membrane were characterized as having one or more chromosomes at the nuclear periphery in the two timepoints following NE breakdown.

## DNA damage assay and quantification of γH2AX foci

U2OS parental and SUN2 Site 2 $^A$ cells were seeded (7.5 × 10$^4$ cells) in Dulbecco's Modified Eagle's Medium (DMEM) supplemented with Pen/Strep onto round cover slips (16 mm, #1) in 35 mm tissue culture-treated dishes. SUN2 Site 2$^A$ expression was induced 24 hr before IR with the addition of Doxycycline to a final concentration of 1 μg/ml. A total dose of 5 Gy IR was performed to induce DNA double strand breaks. Cells were further incubated in doxycycline-containing media for 1, 4, 8, and 24 hr. Control cells not exposed to IR were harvested after 24 hr of doxycycline induction. Cells were fixed with 4% PFA (methanol-free) for 10 min at RT, followed by three PBS washes. Fixed cells were then permeabilized with 0.2% Triton X-100 in PBS for 10 min, followed by three PBS washes. Fixed cells were blocked with PBS/10% FBS for 2 hr at 4°C. Primary antibody (Anti-γH2AX) were incubated overnight at 4°C in PBS/10% FBS (1:1000) in home-made humidified chambers, followed by three times wash with PBS/0.1% Triton X-100 for 5 min. Secondary antibody was prepared in PBS/0.15% FBS, and cells were incubated with secondary antibody at room temperature for 2 hr in the humidified chambers, followed by three times wash with PBS/0.1% Triton X-100 for 5 min and one 5 min wash with PBS. Cover slips were then incubated in DAPI-containing PBS for 5 min, washed three times with PBS, mounted onto glass slide in non-hardening mounting media and sealed with nail polish.

Quantification of H2AX signal was performed using ImageJ (bundled with Java 1.8.0_172) software. Each nucleus was masked and the number of spots within the mask with size between 0.4 and 3 μm was counted as foci. More than 90 parental and SUN2 Site 2 $^A$ cells in each timepoint from three independent experiments were analyzed, and statistical analysis was performed using one-way analysis of variance method on GraphPad Prism v9.0.

## Generation CRISPR/Cas9-mediated gene knockout cells

Cells were transfected with single biscistronic gRNA-Cas9 plasmid at 80% confluency with Mirus LT1 transfection reagent in 24-well tissue culture plate format. Forty-eight hours later, cells were trypsinized and expanded into a 10-cm tissue culture dish in the presence of 2 μg/ml puromycin for 72 hr. Cells were maintained in fresh media (without puromycin) for additional 72 hr before validation by western blotting. Knockout clonal cell lines were generated by single cell sorting using fluorescent-assisted cell sorting (FACS).

## Genome-wide CRISPR/Cas9 screen

The TKOv3 CRISPR/Cas9 library was a gift from Jason Moffat (Addgene #90294). The sgRNA library and second generation lentiviral packaging plasmids were transfected into HEK293T cells. Virus titre was determined for optimizing transduction at MOI of 0.3 in HEK TF SUN2 Site 2$^D$ cells in T175 tissue culture flask format. For the screen, at least 125 × 10$^6$ HEK TF SUN2 Site 2$^D$ cells were infected in T175 flasks to achieve a 250-fold coverage of the library after selection at day 1. On day 2, fresh media was replaced. On day 3, cells were selected with 2 μg/ml puromycin (Gibco) for 72 hr. Cells were seeded into two technical replicates with at least 25 × 10$^6$ total cells per replicate in T175 flasks. Twenty-four hours prior to day 7 sorting, 100 ng/ml doxycycline (Sigma-Aldrich) was added to the cell medium to induce the expression of the transgene SUN2 Site 2$^D$ transgene. On the day of sorting, cells were trypsinized and collected into a single flask. From this, 25 × 10$^6$ reference cells were collected, pelleted and frozen. In addition, two replicates of 25 × 10$^6$ cells were further seeded for day 13 sorting. The rest of the cells were subjected to FACS and 1% of the brightest GFP cells was collected using a BD FACSAria3 sorter. Transgene induction, reference population collection, sorting, and seeding for day 17 were repeated as described above. Genomic DNA was extracted from each cell population using a QIAGEN BloodMaxi kit (for reference samples) or BloodMini kit (for sorted samples) according to the manufacturer's protocol. sgRNAs were PCR amplified from the entire isolated genomic DNA using NEBNext Ultra II Q5 Master Mix (NEB) and the primers v2.1-F1 and v2.1-R1, according to the TKOv3 protocol. PCR reactions were pooled again, after which a second PCR was performed to attach indices and sequencing adapters using the primers i5 and i7. The PCR reaction was loaded onto a 2% agarose gel, the 200 bp band excised and purified using a GeneJet PCR Purification kit (Thermo Fisher Scientific). Concentration and quality of the purified bands were assessed by using TapeStation

and pooled library at 10 nM in 30 µl were prepared. The library was analyzed by deep sequencing on an Illumina NextSeq500 at the Oxford Genomics Centre. Gene rankings were generated using the MAGeCK algorithm.

## Flow cytometry

Cells were trypsinized and resuspended in FACS buffer (2% FCS, 1 mM Ethylenediaminetetraacetic acid (EDTA) in PBS). Cells were analyzed using BD LSRFortessa X-20 flow cytometer on plate reader standard mode.

## Acknowledgements

We thank Z Ji for preliminary experiments with SUN2 truncations, M Gullerova, R Ketley, A Alagia, Q Long, and F Esashi for advice on the DNA damage experiments, V D'Angiolella for reagents, L Witty for help with high-throughput sequencing, A Wainman for assistance with microscopy, the Dunn School Flow Cytometry Facility for help with cell sorting and S Bahmanyar for discussions. P Carvalho is supported by an ERC Consolidator grant (GA: 817708) and an investigator award from The Wellcome Trust (223153/Z/21/Z).

## Additional information

### Funding

| Funder | Grant reference number | Author |
|---|---|---|
| European Research Council | 817708 | Pedro Carvalho |
| Wellcome Trust | 223153/Z/21/Z | Pedro Carvalho |
| Cancer Research UK | Discovery Programme DRCNPG-Nov21\100004 | Ulrike Gruneberg |
| Medical Research Council | MR/K006703/1 | Ulrike Gruneberg |
| Edward Penley Abraham Fund | RF 280 | Ulrike Gruneberg |

The funders had no role in study design, data collection and interpretation, or the decision to submit the work for publication. For the purpose of Open Access, the authors have applied a CC BY public copyright license to any Author Accepted Manuscript version arising from this submission.

### Author contributions

Logesvaran Krshnan, Conceptualization, Formal analysis, Validation, Investigation, Visualization, Writing – original draft, Writing – review and editing; Wingyan Skyla Siu, Michael Van de Weijer, Daniel Hayward, Formal analysis, Validation, Investigation, Visualization, Writing – review and editing; Elena Navarro Guerrero, Formal analysis, Investigation, Writing – review and editing; Ulrike Gruneberg, Supervision, Validation, Visualization, Writing – review and editing; Pedro Carvalho, Conceptualization, Formal analysis, Supervision, Funding acquisition, Writing – original draft, Project administration, Writing – review and editing

### Author ORCIDs

Logesvaran Krshnan ![ORCID] http://orcid.org/0000-0001-6281-1587
Michael Van de Weijer ![ORCID] http://orcid.org/0000-0002-0954-0228
Pedro Carvalho ![ORCID] http://orcid.org/0000-0002-9691-5277

### Decision letter and Author response

Decision letter https://doi.org/10.7554/eLife.81573.sa1
Author response https://doi.org/10.7554/eLife.81573.sa2

## Additional files

### Supplementary files
• Transparent reporting form

### Data availability
Sequencing data have been deposited European Nucleotide Archive repository and have the accession number PRJEB54102.

The following dataset was generated:

| Author(s) | Year | Dataset title | Dataset URL | Database and Identifier |
|---|---|---|---|---|
| Krshnan L, Siu WS, van de weijer M, Hayward D, Guerrero EN, Gruneberg U, Carvalho P | 2022 | Genome-wide CRISPR-Cas9 screen (TKOv3) data on SUN2 degradation | https://www.ebi.ac.uk/ena/browser/view/PRJEB54102 | ENA, PRJEB54102 |

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
