## [Editor Report]

This important paper describes a "ERAD-like" pathway for turnover of the SUN2 protein. Here, ubiquitylation of SUN2 in the nucleoplasm by the SCFbTRCP ubiquitin ligase leads to extraction of the membrane protein by p97 for delivery to the proteasome in a process that is regulated by the CK2 kinase and the CTDNEP1 phosphatase. Non-degradable forms of SUN2 promote altered nuclear architecture and a delay in double strand break repair. The conclusions are based on strong biochemical and cell biological data, and are supported by multiple types of genetic approaches.

---

## [Decision Letter]

**Decision letter after peer review:**

Thank you for submitting your article "Regulated Degradation of the Inner Nuclear Membrane Protein SUN2 Maintains Nuclear Envelope Architecture and Function" for consideration by *eLife*. Your article has been reviewed by 3 peer reviewers, one of whom is a member of our Board of Reviewing Editors, and the evaluation has been overseen by Vivek Malhotra as the Senior Editor. The following individual involved in the review of your submission has agreed to reveal their identity: Yihong Ye (Reviewer #3).

Essential revisions:

Overall the reviewers feel that the paper is strong and relatively complete. However, there are two aspects that we feel would improve the paper.

1) Although we realize FBXW11 (b-TRCP2) was identified in the CRISPR screen, we feel a formal genetic demonstration of the involvement of FBXW11 and/or b-TRCP1 as a redundant backup enzyme is important, given the conclusions. The extensive use of the degron mutants and binding, while providing important insight, isn't really sufficient genetic evidence. We also realize that making KOs of these E3s is challenging and they are likely lethal as a DKO, but given the wide use in the field of shRNAs for reducing their levels, the use of shRNAs for this experiment would be ok. Essentially, we are asking for further validation of the genetic screening data.

2) A more complicated question concerns the extent to which this pathway works on endogenous SUN2. In available protein turnover data (PMID: 34626566), SUN2 is really not an unstable protein (half-life of 8h or more). This could reflect that most of the protein is assembled into a stable complex, but of course, there could be small amounts of protein that didn't assemble properly which could be degraded by this pathway. In the experiments reported, it is likely that significant levels of SUN2 are not associated with partner proteins and is therefore mimicking a potential QC pathway. So we are wondering if you can measure turnover of the endogenous protein in the context of b-TRCP shRNA and see if you detect any turnover of the protein normally. If not, the reviewers feel that the paper/data itself is still valuable but that it might require re-working the paper a bit to more of a quality control angle. One experiment that could shed light on this question is to reduce the levels of SUN2-associated proteins and see if that increases the pool of SUN2 that is degraded in a b-TRCP-dependent manner. One could use MLN4924 to do prescreens to determine which subunits of the SUN2 complex might be most critical for the maintenance of its stability. It is also possible you have considered this model and have data for or against this model, and if so, this could change how to address this point.

3. There are several other suggestions for experiments or changes in the text that will improve the paper.

*Reviewer #3 (Recommendations for the authors):*

1. Although it is understandable that for mutagenesis and CRISPR screen studies, over-expressed SUN2 bearing a tractable tag is necessary, it is important for the authors to validate key findings using endogenous SUN2. In this regard, the role of CK2 (Figure 5F) and SCF β TrCP (Figure 5D) in SUN2 turnover in phosphatase CTDNEP1 knockout cells should be further confirmed with endogenous SUN2.

2. To address the physiologic relevance of the reported SUN2 degradation, the authors may consider genetically editing the site 2 sequence of SUN2 to abolish this regulation. They can then examine whether this can cause any cellular defects as shown in Figure 6. If this is technically challenging, they need to carefully discuss the limitations of their experimental design and include alternative interpretations in the Discussion section.

3. They should consider changing the title and discuss the potential physiological conditions that may trigger SUN2 degradation.

4. The authors should also comment on the subcellular localization of CK2 and CTDNEP1 and whether that can support the degradation of inner nuclear envelope proteins.

---

## [Author Response]

Essential revisions:Overall the reviewers feel that the paper is strong and relatively complete. However, there are two aspects that we feel would improve the paper.1) Although we realize FBXW11 (b-TRCP2) was identified in the CRISPR screen, we feel a formal genetic demonstration of the involvement of FBXW11 and/or b-TRCP1 as a redundant backup enzyme is important, given the conclusions. The extensive use of the degron mutants and binding, while providing important insight, isn't really sufficient genetic evidence. We also realize that making KOs of these E3s is challenging and they are likely lethal as a DKO, but given the wide use in the field of shRNAs for reducing their levels, the use of shRNAs for this experiment would be ok. Essentially, we are asking for further validation of the genetic screening data.

We thank reviewers for the suggestion to provide further genetic evidence of the involvement of βTrCP1 and 2 F-box proteins in the degradation of SUN2. We now show that maximum stabilization of endogenous (Figure Supplement S4D) and transgenic (Figure S2) SUN2 is observed upon simultaneous depletion of βTrCP1 and βTrCP2 indicating that these F-Box proteins are redundant. Depletion of βTrCP1 alone did not impact SUN2 levels while depletion of βTrCP2 increased SUN2 steady state levels, with the effect being more pronounced for overexpressed SUN2. Depletion of other F-Box proteins did not affect SUN2 levels indicating that the effect observed for βTrCP1 is specific (Figure S2B). These results are in line with the results of our genome wide screen (Figure 4 and S3) and the literature. The differences in the effects of βTrCP1 and βTrCP2 depletion likely result from the relative abundance of the two F-Box proteins in the HEK cells used in this study

2) A more complicated question concerns the extent to which this pathway works on endogenous SUN2. In available protein turnover data (PMID: 34626566), SUN2 is really not an unstable protein (half-life of 8h or more). This could reflect that most of the protein is assembled into a stable complex, but of course, there could be small amounts of protein that didn't assemble properly which could be degraded by this pathway. In the experiments reported, it is likely that significant levels of SUN2 are not associated with partner proteins and is therefore mimicking a potential QC pathway. So we are wondering if you can measure turnover of the endogenous protein in the context of b-TRCP shRNA and see if you detect any turnover of the protein normally. If not, the reviewers feel that the paper/data itself is still valuable but that it might require re-working the paper a bit to more of a quality control angle. One experiment that could shed light on this question is to reduce the levels of SUN2-associated proteins and see if that increases the pool of SUN2 that is degraded in a b-TRCP-dependent manner. One could use MLN4924 to do prescreens to determine which subunits of the SUN2 complex might be most critical for the maintenance of its stability. It is also possible you have considered this model and have data for or against this model, and if so, this could change how to address this point.

Consistent with the published literature, we observe that endogenous SUN2 is normally a long-lived protein. We now show that depletion of the phosphatase CTDNEP1 dramatically reduces the half-life of endogenous SUN2 (Figure 5C). Importantly, the degradation of endogenous SUN2 is inhibited in cells lacking βTrCP (Figure S4D) or treated with MLN4924, CB5083 or Bortezomib (Figure S4C). Therefore, both endogenous and transgenic SUN2 follow the same degradation pathway.

The physiological condition(s) leading to SUN2 degradation remain under investigation as a parallel project in our lab and in our opinion fall outside of the scope of the current work. However, data in the literature and our preliminary results argue against SUN2 degradation being related with its assembly state. SUN2 variants lacking the luminal SUN domain (for example SUN2(1-260)) and therefore unable to assemble with Nesprins (via the SUN domain) into a LINC complex display steady state levels comparable to WT SUN2 (see for example PMID: 20551905). These data were reproduced independently in our lab. In agreement with these observations, our preliminary data indicate that SUN2 levels are unaltered by depletion of Nesprin 1 or 2 (Author response image 1). Thus, assembly with Nesprins does not appear to determine SUN2 stability.

**Author response image 1. sa2fig1:** CRISPR/Cas9-mediated depletion of Nesprin1 (SYNE1) and Nesprin2 (SYNE2) in SUN2 WT or SUN2 2^D^-expressing HEK TF cells.

On the other hand, we observe that the turnover of endogenous SUN2 is influenced by the turnover rates of overexpressed SUN2. This is mostly obvious for cells overexpressing the SUN2 Site 2^A^ and Site 2^D^ mutants, which increase and decrease the stability of endogenous SUN2, respectively (see Figure 1E). Since SUN2 functions as an homotrimer, this effect likely reflects the assembly of Site 2^A^ and Site 2^D^ mutants with endogenous SUN2. We comment on this observation in the Results section.

In summary, our current data strongly support a model in which SUN2 stability is mainly dependent on the phosphorylation state of nucleoplasmic site 2 and appears independent of its assembly state. Since this is at the moment not fully resolved point, it will not be discussed in further detail in the manuscript.

3. There are several other suggestions for experiments or changes in the text that will improve the paper.

These suggestions are addressed point-by-point below.

Reviewer #3 (Recommendations for the authors):1. Although it is understandable that for mutagenesis and CRISPR screen studies, over-expressed SUN2 bearing a tractable tag is necessary, it is important for the authors to validate key findings using endogenous SUN2. In this regard, the role of CK2 (Figure 5F) and SCF β TrCP (Figure 5D) in SUN2 turnover in phosphatase CTDNEP1 knockout cells should be further confirmed with endogenous SUN2.

The original submission showed that depletion of CTDNEP1 lead to strong reduction of endogenous SUN2 levels (Figure 5A-B). Importantly this could be rescued by re-expression of WT CTDNEP1 but not by phosphatase dead CTDNEP1 (Figure 5E). In the revised manuscript, we include new data showing that the turnover of endogenous SUN2 is dramatically accelerated in cells lacking CTDNEP1 (Figure 5C) and that this acceleration is curbed by MLN4924, CB5083 and Bortezomib (Figure supplement 4C). Together with the additional data, these experiments demonstrate that CTDNEP1 regulates the turnover of both overexpressed and endogenous SUN2.

2. To address the physiologic relevance of the reported SUN2 degradation, the authors may consider genetically editing the site 2 sequence of SUN2 to abolish this regulation. They can then examine whether this can cause any cellular defects as shown in Figure 6. If this is technically challenging, they need to carefully discuss the limitations of their experimental design and include alternative interpretations in the Discussion section.

We thank the reviewer for the suggestion. It would be great to gene edit the SUN2 locus to introduce the desired mutations. But as pointed out this is not trivial, in particular considering that the desired mutations would need to be introduced in both chromosomal copies.

3. They should consider changing the title and discuss the potential physiological conditions that may trigger SUN2 degradation.

Please see our comments to point 2 above. We believe that the title is appropriate.

4. The authors should also comment on the subcellular localization of CK2 and CTDNEP1 and whether that can support the degradation of inner nuclear envelope proteins.

We thank the reviewer for the suggestion. We now mention that SCF^βTrCP^ ubiquitin ligase, CK2 and CTDNEP1 all localize to the nucleus there by being able to access SUN2.